# MCHeart: Multi-Channel-Based Heart Signal Processing Scheme for Heart Noise Detection Using Deep Learning

**DOI:** 10.3390/biology12101291

**Published:** 2023-09-27

**Authors:** Soyul Han, Woongsun Jeon, Wuming Gong, Il-Youp Kwak

**Affiliations:** 1Department of Applied Statistics, Chung-Ang University, Seoul 06974, Republic of Korea; soyul5458@cau.ac.kr; 2School of Electrical and Electronics Engineering, Chung-Ang University, Seoul 06974, Republic of Korea; wjeon@cau.ac.kr; 3Lillehei Heart Institute, University of Minnesota, Minneapolis, MN 55455, USA; gongx030@umn.edu

**Keywords:** heart murmur detection, biological signals, feature extraction, smart healthcare, light CNN, multiple attention network, deep learning

## Abstract

**Simple Summary:**

Cardiovascular disease is a major global health concern. Early detection is vital, with phonocardiograms (PCGs) offering valuable heart sound data, including murmurs. Research automating PCG analysis is growing, addressing challenges like the 2022 PhysioNet Challenge. Our innovation, the MCHeart system, focuses on irregular heart murmurs, combining S1/S2 features, smoothing, and a residual LCNN architecture with multi-head self-attention for enhanced feature extraction.

**Abstract:**

In this study, we constructed a model to predict abnormal cardiac sounds using a diverse set of auscultation data collected from various auscultation positions. Abnormal heart sounds were identified by extracting features such as peak intervals and noise characteristics during systole and diastole. Instead of using raw signal data, we transformed them into log-mel 2D spectrograms, which were employed as input variables for the CNN model. The advancement of our model involves integrating a deep learning architecture with feature extraction techniques based on existing knowledge of cardiac data. Specifically, we propose a multi-channel-based heart signal processing (MCHeart) scheme, which incorporates our proposed features into the deep learning model. Additionally, we introduce the ReLCNN model by applying residual blocks and MHA mechanisms to the LCNN architecture. By adding murmur features with a smoothing function and training the ReLCNN model, the weighted accuracy of the model increased from 79.6% to 83.6%, showing a performance improvement of approximately 4% point compared to the LCNN baseline model.

## 1. Introduction

Cardiovascular disease is a major health issue and a leading cause of death worldwide. Despite advances in medical technology, statistics showing high mortality rates from heart disease indicate that it remains a formidable threat to our health. Early detection of symptoms of cardiovascular disease is cost-effective, as it makes treatment easier and more efficient [1]. The health of the heart can be assessed by obtaining information on cardiac disorders through electrocardiograms (ECGs) and phonocardiograms (PCGs). While ECGs provide information about the electrical signals around the heart, PCGs provide information about the acoustic signals produced by the vibrations of heartbeats. Together, these tests help doctors diagnose a wide range of heart conditions. PCGs especially are a cheap and simple non-invasive test that pose no burden on people visiting hospitals for disease prevention. Additionally, PCGs provide important information for diagnosing cardiac diseases [2]. PCGs may include a range of sounds beyond the normal heart sounds, including heart murmurs. Murmurs, which can occur both inside and outside the heart, are typically caused by turbulent blood flow and can be either benign or abnormal/pathological in nature [2]. To accurately judge murmurs from recorded heart sounds, an expert with much clinical experience with various types of murmurs is required.

However, the interpretation of cardiovascular diseases can be somewhat subjective, as it may lead to clinical disagreements among medical practitioners based on years of experience and expertise, resulting in a lack of objectivity [3,4,5]. In recent years, there has been a growing trend in research of using PCG data to automatically detect heart diseases. Pedro et al. (2020) [6] applied an empirical wavelet transform to PCG signal data for preprocessing and then used different machine learning models for classification, including the support vector machine and k-nearest neighbor models. Banerjee and Majhi (2020) [7] proposed a deep learning model for noise detection using mel-frequency cepstral coefficient features. Boulares et al. (2021) [8] also proposed a heart disease recognition model using unsupervised and supervised learning methods based on convolutional neural networks (CNNs).

The theme of the George B. Moody PhysioNet Challenge for 2022 was Heart Murmur Detection from PCG Recordings [9]. The competition aimed to identify the presence or absence of murmurs and normal vs. abnormal clinical outcomes from heart sound recordings collected from multiple auscultation locations using a digital stethoscope. The competition is beneficial because congenital and acquired heart diseases affect many children in underprivileged countries where early diagnosis is difficult due to the lack of infrastructure and cardiology specialists. The Challenge required participants to design and implement a working open-source algorithm that can determine the presence of murmurs and identify the clinical outcomes from recordings and demographic data. A total of 87 teams submitted 779 algorithms during the Challenge. Lu et al. (2022) [10] used a lightweight CNN and a random forest model to detect heart murmurs and classify clinical outcomes, achieving 1st and 10th place in the challenge tasks for murmur detection and clinical outcome classification, respectively. McDonald et al. (2022) [11] used a recurrent neural network and hidden semi-Markov model approach to detect heart murmurs in PCG recordings, applying multiple hidden semi-Markov models to produce multiple segmentations of the signal and compare their confidence and then classifying murmurs and producingrobust segmentations. This model ranked second in the murmur detection task. Lee et al. (2022) [12] proposed a deep learning-based model using a log-mel spectrogram and light CNN (LCNN) to identify heart murmurs from a PCG.

Most existing models use only spectrogram features extracted from raw data to capture time–frequency characteristics. However, in addition to the spectrogram features, we utilize background knowledge about heart signals. Through this knowledge, we can divide the heart sounds into S1 and S2 components and their complements (S1 and S2 are the two distinct sounds produced by the heart during each heartbeat). Inspired by irregular heart murmurs, we propose a multi-channel-based heart signal processing (MCHeart) system, a heart murmur detection system that addresses irregular heart murmurs using the following three components: (1) We focus on utilizing additional feature information from the cardiac activity sounds, such as S1 and S2, derived from PCG data. By extracting cardiac activity sound features and applying mel spectrograms, we obtain richer temporal and frequency domain characteristics. (2) A smoothing function is applied to minimize the noise or irregularity of the heart sound signal. (3) Additionally, we introduce the residual LCNN (ReLCNN) architecture by incorporating residual blocks and multi-head self-attention (MHA) into the model proposed by Lee et al. (2022) [12]. The baseline model proposed by Lee et al. (2022) [12] (CAU_UMN team) achieved notable rankings in the PhysioNet Challenge 2022. In both the murmur detection and clinical outcome detection categories, this model secured the fifth position among a total of 87 participating teams [9]. We aimed to further enhance this model with our own insights.

Utilizing the ReLCNN model resulted in an enhancement of weighted accuracy from 79.6% to 80.5%. Upon incorporation of S1 and S2 components into the ReLCNN model, the weighted accuracy further improved to 82.0%. Moreover, the application of a smoothing function led to an increased accuracy of 83.7%.

## 2. Methods

The overview of the ReLCNN model proposed in this study is shown in Figure 1. As shown in the feature extraction section in the figure, various preprocessing features, such as peak intervals, S1 and S2, murmurs, and envelopes, in addition to the commonly used spectrogram feature, are employed with the raw PCG audio signal. For data augmentation, we used mixup and cutout techniques [13,14]. For the model, we employed the ReLCNN architecture, which is an extension of the LCNN model proposed by Lee et al. (2022) [12] with the addition of residual connections and MHA.

We describe the features extracted using PCG data in Section 2.1 and the models trained to detect murmur abnormalities in Section 2.2.

### 2.1. Feature Extraction from PCGs

We hypothesized that the accuracy of murmur detection models that can determine the current state of the heart will increase as the feature information in phonocardiography becomes more diverse. We conducted research using four features: spectrograms, S1 and S2, the complements of S1 and S2, and envelopes. We transformed the one-dimensional PCG data into a two-dimensional spectrogram and used it as input to the CNN. Spectrograms provide important diagnostic information related to the cardiac state that is not clearly visible in the temporal domain of one-dimensional data by visually representing the frequency and temporal components of the signal, thereby improving the accuracy of prediction algorithms.

#### 2.1.1. Log-Mel Spectrogram

The log-mel spectrogram is a feature extraction technique that takes into account human auditory characteristics. It performs a short-time Fourier transform (STFT) and applies mel-scale filters to obtain the power for each frequency band, which is then converted to a logarithmic scale. In order to determine what information is present in the original signal, the signal must be decomposed into its frequency components using a Fourier transform. However, applying a Fourier transform to the entire signal would result in the loss of temporal information and only frequency information would remain. Therefore, to preserve the temporal information, the signal needs to be divided into short time intervals, which is known as the STFT. The number of data points sampled per second in the signal data is called the “sampling rate”, and the process of dividing time units into short intervals is called “framing”. To preserve the time information, the signal is divided into small pieces, with the size of each piece referred to as the “window size”. However, since the window is defined with the edges cut off, the result is similar to the sound being truncated. Therefore, to end naturally, a Hamming window is applied to each frame to give weight to the center where more information can be seen. However, since the application of the Hamming window results in the loss of information at the end of the frame, a Fourier transform needs to be applied with overlapping frames to prevent the loss of information at the edge of the frame. The degree to which certain intervals overlap is called the “hop length”, and the STFT spectrogram is generated based on the window size and hop length. The mel spectrogram [15] is calculated by converting the power for each frequency band of the generated STFT spectrogram to the mel scale using Equation (Equation 1),
(1)$$\begin{array}{cc} \; & Mel\left(f\right)=2595\;log_{10}\left(1+\frac{f}{700}\right) \end{array}$$
(2)$$\begin{array}{cc} \; & LogMel\left(f\right)=10\;log_{10}\left(\frac{Mel(f)}{ref}\right) \end{array}$$
where *f* represents frequency. The log-mel spectrogram is obtained by taking the logarithm of the power of the mel spectrogram and converting it to dB using Equation (2), and ref is a reference value that Mel(f) is relatively scaled to.

#### 2.1.2. Peak Interval

Peaks refer to the areas in a signal where the amplitude or energy is high. This is an important clue for inferring the characteristics of sound. In this paper, the peak interval is defined as the distance between peaks in the PCG signal. Numerous medical studies and previous PhysioNet competitions have utilized the R-R interval, which represents the distance between peaks in an ECG signal, as a tool for evaluating anomalies in cardiac disease [16,17,18]. Since the R-R interval in an ECG effectively represents heart rate variability (HRV) [17,18], an additional function is required for PCGs to express HRV. Thus, the “peak interval”, which corresponds to the R-R interval, was devised for representing HRV in PCG signals.

Patients have heart murmurs between systole and diastole, which generate waveforms corresponding to the noise. Therefore, we hypothesized that patients with heart disease would have more peaks due to abnormal heart murmurs between systole and diastole. More peaks result in shorter peak intervals. To confirm this hypothesis, we compared the average peak interval between subjects with murmurs and those without murmurs in the PhysioNet Challenge 2022 dataset. We found that the interval was approximately 49% longer in the absence of murmurs, as illustrated in Figure 2. Patients with murmurs exhibited a higher number of peaks during the same time period compared to those without murmurs. Peak detection can be easily performed on raw audio using the “ecg_peaks” function from the Python library “Neurokit2” [19]. Although we intended to use the peak interval value as a sequence, it was difficult to calculate the exact peak interval due to noise, so we used the average peak interval instead of the sequence form.

#### 2.1.3. Boundary Detection for Fundamental Heart Sounds and Heart Murmurs

The heart sound is a mechanical activity signal of the heart. The sound produced by the fundamental heart sounds (FHSs) consists of two components, S1 and S2, as illustrated in Figure 3. S1 and S2 are the most basic heart sounds, while heart murmurs are sounds occurring during both systole and diastole. The presence of heart murmurs can be indicative of cardiac issues, although they may also be occasionally heard in healthy children and young adults (Akkarapol et al., 2012 [20]). FHSs and heart murmurs are essential elements when analyzing PCG signals to diagnose various heart diseases. However, in the PhysioNet Challenge 2022 dataset, FHS information is not provided, resulting in the lack of crucial characteristic information to judge the heart’s condition. To improve the performance of the model in automatically identifying heart diseases, FHS feature information would be necessary. Therefore, we referred to the time–frequency-domain approach proposed by Ghosh and Ponnalagu (2019) [21] for automated FHS detection using PCG signals. The boundaries of the heart cycle, S1, S2, and the boundaries of systole and diastole murmurs contained in the extracted heart cycle were detected. We then applied a log-mel spectrogram to each detected boundary signal to extract spectrogram features for identifying heart diseases.

The process of detecting FHS boundaries and applying spectrograms was performed in six steps: (1) amplitude normalization, (2) applying a Butterworth low-pass filter, (3) creating a PCG signal envelope, (4) selecting a threshold, (5) detecting boundaries, and (6) applying a log-mel spectrogram. This process is demonstrated in Figure 4.

First, according to Ghosh et al. (2019) [21], HRV signals vary in amplitude depending on patient factors such as physiology, gender, and age, making it difficult to predict the dynamic range of the signal. Therefore, normalization was applied to the HRV signal data to transform the range of amplitudes to between −1 and 1. The signal data were normalized according to Equation (Equation 3) below, where i=1,2,3,4,…,N, and *N* represents the total number of samples. $Ns_{i}$ represents the normalized signal.
(3)$$\begin{array}{cc} \; & Ns_{i}=\frac{s_{i}}{|s_{i}{|}_{max}} \end{array}$$

Second, to eliminate high-frequency components, we preprocessed the normalized signal using a low-pass Butterworth filter with a 150 Hz cutoff frequency [21]. This cutoff frequency, a hyperparameter, was selected after considering the trade-off between effectively reducing high-frequency noise and preserving vital cardiac sound information. Ghosh et al. (2019) [21] applied the Stockwell transform to remove heart murmurs. In this study, heart murmur information was an important factor for classifying normal and abnormal heart sounds. Therefore, the Stockwell transform was not performed.

Third, since the sampling frequency of the PCG signal is 4000 Hz, the variation of the frequency amplitude in the time domain is fast. Rapid-amplitude variation makes it difficult to detect the onset and end of the heart sounds S1 and S2. Therefore, it is necessary to detect the envelope of the PCG signal to minimize this variation [22]. The envelope was extracted using the *signal.hilbert* function of the scipy library. The graphical representation of the extracted envelope is shown in step three in Figure 4. Since the PCG signal is almost symmetric around the horizontal axis of zero, we considered only the positive part of the signal for computational efficiency. Fourthly, in order to automatically detect the boundaries of cardiac sound activity, the threshold needs to be determined. Therefore, selecting an appropriate threshold value ($Th_{sh}$) is one of the most important tasks. In this study, Equation (Equation 4) was used to find an appropriate threshold value, $Th_{sh}$.
(4)$$\begin{array}{cc} \; & Th_{sh}=\frac{1}{N}\sum En_{i}+\frac{1}{N}\sum {\left(En_{i}-\mu \right)}^{2} \end{array}$$
where *N* represents the total number of samples. $En_{i}$ is the signal envelope. The fourth step in Figure 4 shows the FHS boundary detection, represented by red dashed lines, based on the calculated threshold value.

Fifth, the boundary detection was separated using the previously calculated $Th_{sh}$, which represents the detection of S1 and S2, as well as the boundaries of the systole and diastole. In Figure 4, step five involves the process of separating the FHS boundary (represented by the red dashed lines) based on the threshold calculated in step four. The FHS boundary values of 1 correspond to the normal heart sounds (namely, S1 and S2), while values of 0 represent features associated with noise occurring during systole and diastole. Sixth, the log-mel spectrogram was applied to the two separated signal envelope features. Step six in Figure 4 shows the log-mel spectrogram of the separated boundaries. The 2D image of the log-mel spectrogram was used as the input for the experimental model.

Figure 5 shows new features considered in this study. The first and second columns represent the spectrograms of S1 and S2 and of systolic and diastolic murmurs, respectively, obtained from step six shown in Figure 4. The third column shows the log-mel spectrogram features extracted from the envelope feature in step three shown in Figure 4.

#### 2.1.4. Application of Smoothing Method

Smoothing is a signal processing technique used to reduce noise and random irregularities in a signal, aiming to make it smoother [23]. For example, in the context of stock price data, smoothing by noise removal facilitates the identification of trends. In practice, the moving-average filter is commonly employed to achieve signal smoothing, calculating the average for data points within a specified time window to create a smoother representation; it is often used for trend or pattern analysis.

**Figure 5 biology-12-01291-f005:**
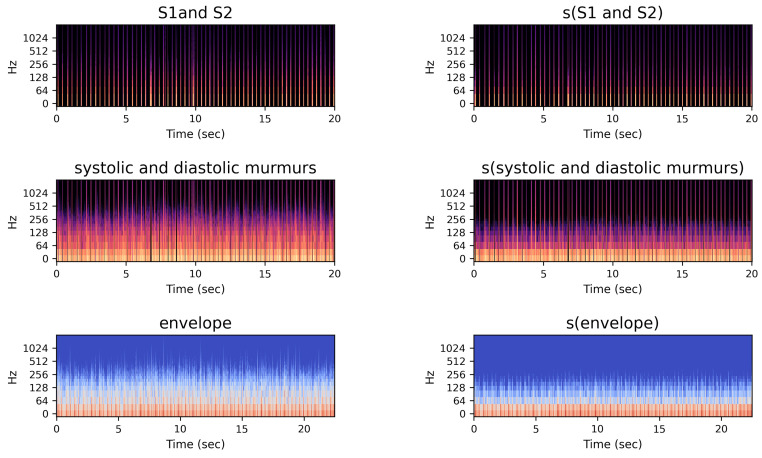
Spectrogram images of proposed multi-channel features.

The right column of Figure 5 shows the log-mel spectrograms applied after the smoothing method. We performed signal smoothing using a moving-average filter. The moving-average filter is represented by the window array, which contains 1s and is divided by the window size to normalize the filter. Let x[n] be the input signal and w[m] be a moving-average filter window of size *N*, where m=0,1,…,N−1. The smoothed signal y[n] at index *n* is given by the convolution operation:(5)$$y\left[n\right]=\sum_{N-1}^{m=0}x\left[n-m\right]w\left[m\right].$$

### 2.2. Residual LCNN (ReLCNN) Model

In this study, we employed the system proposed by Lee et al. (2022) [12] as the underlying framework. Lee et al. (2022) [12] applied an LCNN model to PCG data. We propose a modified version of the LCNN model called ReLCNN, which incorporates residual blocks, activation functions, and multi-head self-attention mechanisms. The multi-head self-attention mechanisms were adopted from the winning architecture of the 2021 PhysioNet Challenge [24].

The LCNN model is a CNN-based model. It is widely used in the speech domain and has already demonstrated good performance in various speech competitions, including ASVspoof 2017, 2019, and 2021 [25,26]. The LCNN framework, proposed by Xiang et al. (2018) [27], consists of three models: LCNN-4, LCNN-9, and LCNN-29. In this study, we modified the LCNN-9 model. The LCNN-9 model comprises five convolutional layers and four network-in-network (NIN) layers. The NIN layer, introduced by Lin et al. (2014) [28] for classification tasks, offers a novel deep neural network architecture. The LCNN-9 model integrates max-feature-map (MFM) layers into the NIN layers to perform feature selection between convolutional layers and reduces the number of parameters by utilizing max pooling.

In this study, the ReLCNN model proposed is a modified version compared to the LCNN-9 model. It includes deeper layers and selectively employs batch normalization along with the Swish activation function. The model architecture presented in this paper is depicted in Figure 6. The model employs convolutional filters with sizes of 32, 32, 48, 48, 64, 64, 32, 32, and 32. The kernel size for the first convolutional layer is set to 5, while the remaining convolutional layers have kernel sizes of 1 or 3. The LCNN block, which constructs the LCNN model, is depicted in Figure 7a and consists of five parameters (f,k,m,b,a). *f* and *k* represent the parameters for the filter and kernel sizes of the convolutional layer, respectively. *m* indicates the usage of max pooling, *b* represents the usage of batch normalization, and *a* denotes the usage of the Swish activation function. The dotted boxes within the LCNN block represent optional applications, indicating that the corresponding parameter is selectively applied when it is set to 1.

#### 2.2.1. Residual Block

In the case of CNNs, as the depth of the layers increases, the problem of vanishing gradients arises. This means that, as the depth of the network increases, the information from the hidden layers closer to the input tends to fade away, leading to difficulties in effective learning. To overcome this issue of vanishing gradients, He et al. (2016) [29] proposed the residual neural network architecture. The residual neural network architecture consists of multiple residual blocks stacked together. A residual block incorporates a simple yet effective idea, taking the output F(x) after it passes through several layers and directly adding it to the input value *x*. This approach aids in better convergence during training. The fundamental structure of a residual block is illustrated in Figure 7b. In cases where the input and output dimensions differ, max pooling or 1 × 1 convolution is added to the residual connection for adjustment purposes. Our model has three residual connections, as in shown Figure 6.

#### 2.2.2. Activation Function

Activation functions play a significant role in the classification performance of deep neural networks [30,31]. Currently, the rectified linear unit (ReLU) activation function is widely favored for its effectiveness in optimizing models when dealing with positive inputs [32]. The formula for the ReLU activation function is provided in Equation (Equation 6). However, a problem arises when the activation value is 0, as it leads to a situation where all neuron outputs become 0 in the subsequent layer, causing a halt in updates and learning. To overcome this limitation, the Swish activation function was proposed [30]. It is a simple formula that multiplies the input *x* by the sigmoid function, as shown in Equation (Equation 7).
(6)$$\begin{array}{cc} \; & ReLU(x)=max(0,x) \end{array}$$
(7)$$\begin{array}{cc} \; & Swish(x)=x\cdot Sigmoid(\beta x) \end{array}$$
(8)$$\begin{array}{cc} \; & Sigmoid\left(z\right)={\left(1+exp\left(-z\right)\right)}^{-1} \end{array}$$

In the Swish activation function, β represents the trainable parameter and *x* denotes the input data, while the sigmoid function is defined as shown in Equation (Equation 8). According to experimental results, when training deep layers, Swish demonstrates superior image classification performance compared to ReLU owing to its ability to preserve gradients effectively [30,31]. Additionally, Jinsakul et al. (2019) improved the performance of the initial colorectal cancer screening system by modifying the activation function to Swish in the Xception model [33]. In this study, we utilized the Swish activation function at the end of the LCNN block, as depicted in Figure 7a.

#### 2.2.3. Multi-Head Self-Attention

MHA is a fundamental concept in architectures that finds wide applications in sequential models [34]. In this study, we employed MHA where the query, key, and value vectors are input identically to allow the learning of weights on more important parts of the output information. This approach was also utilized by the team that achieved first place in the 2021 PhysioNet Challenge [24].

The idea behind MHA is to use multiple sets of query, key, and value transformations to capture different aspects of the input feature map, enabling the model to focus on different patterns and relationships. By attending to relevant parts of the input feature map, the model can better extract meaningful features and improve overall performance in various tasks.

## 3. Experiments

### 3.1. Dataset

The data used in this paper were from a dataset provided by the George B. Moody PhysioNet Challenge 2022 [9,35]. The organizers of the competition possessed data from 1568 pediatric subjects; however, after the conclusion of the competition, only 942 subjects’ training data were made available, while the validation and test data remained undisclosed. The publicly accessible dataset comprises 3163 PCG signal data samples collected from the 942 patients with a 4000 Hz sampling rate. We received multiple PCG data samples for each patient based on different auscultation locations.

It is important to note that the number of PCG recordings for each patient varies, as do the auscultation locations from which the data were collected. For instance, for patient A, two PCG recordings were collected when auscultating at the aortic valve (AV) and pulmonic valve (PV) locations. On the other hand, patient B underwent auscultation at the AV, PV, tricuspid valve (TV), and mitral valve (MV) locations, resulting in four PCG data recordings. The auscultation locations include the AV, PV, TV, MV, and others (Phc). We excluded the “others” category, which represents non-specific locations, as shown in Figure 8. The four specific locations are known for being effective in the detection of valvular sounds.

The files provided by the competition are of four types (wav, hea, tsv, txt), and their descriptions are as follows:wav: an audio recording file containing heart sounds corresponding to each auscultation location;hea: a header file providing descriptions and metadata for the corresponding wav files;tsv: a tab-separated values file containing information on the start and end positions of heart sounds S1 and S2 for each auscultation location;txt: a patient-specific description file, including demographic information (gender, age group, height, weight, pregnancy status), noise characteristics (timing, shape, pitch, grading, quality, etc.), and heart murmur labels.

The validation and test datasets consisted of wav audio files with recorded heart sounds and txt text files containing only demographic information. Therefore, any other data beyond demographic information and heart sound recordings were unavailable. Table 1 presents the distribution of heart murmur labels in the training dataset. Among the patients, 695 (73.78%) had no heart murmurs (absent), 179 (19%) had a heart murmur (present), and 68 (7.22%) had an unknown heart murmur status. Furthermore, for the 179 patients with a heart murmur, the distribution of the most audible auscultation locations was found to vary across the PV, TV, MV, and AV locations.

As the competition organizers did not provide separate validation and test datasets, we performed direct data splitting using the training dataset for validation purposes. Assuming that the validation data would be composed with a similar ratio of heart murmur labels as the training dataset, we conducted stratified sampling based on the proportion of heart murmur labels. In this paper, we randomly split the training data into an 8:2 ratio for training (80%) and validation (20%) purposes.

### 3.2. Data Augmentation

Data augmentation techniques are employed to generate diverse new samples from existing datasets. These techniques have been used in CNNs to prevent overfitting and improve the generalization performance, thus creating robust models. Given the limited availability of training data in this study, it was crucial to robustly train the model by incorporating a more diverse dataset. Therefore, for audio classification, we applied two widely used techniques, Cutout [14] and Mixup [13], to augment and introduce disturbance to the spectrogram features.

Firstly, Mixup [13] randomly samples two different samples according to weights obtained from the beta distribution and blends them to create a new image. The formula for Mixup is as follows:$$\tilde{x}=\lambda x_{i}+\left(1-\lambda \right)x_{j}$$
$$\tilde{y}=\lambda y_{i}+\left(1-\lambda \right)y_{j},$$
where the parameter λ is extracted from a β(α,α) distribution, taking values between 0 and 1 (λ∈[0,1]), while α can take values between 0 and infinity (α∈ (0, *∞*)). Here, $x_{i}$ and $x_{j}$ represent different input data and $y_{i}$ and $y_{j}$ correspond to their respective one-hot-encoding values. The augmented data $\tilde{x}$ and $\tilde{y}$ are generated by mixing the two data samples based on the λ value extracted from the β(α,α) distribution. We set the α value to 0.5.

Cutout [14] randomly masks contiguous portions of the input image by setting them to zero, creating a new image. This approach helps the model focus on the overall context of the image rather than concentrating on specific features, making it more robust to noisy images. As PCG data contain significant noise, applying data augmentation techniques to the spectrogram images helps prevent overfitting and improve generalization performance. This ensures that the model considers a broader image context when making predictions, leading to a more robust model.

### 3.3. Implementation Details

In this study, we conducted optimization through a hyperparameter search. Table 2 provides a comprehensive list of explored hyperparameters along with the selected values. The log-mel spectrogram was chosen as it demonstrated the highest performance among the spectrogram options. The “trim” is a hyperparameter that determines how much of the original signal to trim from the beginning and the end. When listening to the PCG sound data directly, it was initially hypothesized that the front and back portions might be less important and that trimming these portions from the signal would improve performance. However, in practice, it was observed that the model achieved higher performance when the data were not trimmed. As the sampling frequency of the heart sound data was 4000, a value of 4000 means trimming 1 s of data, while a value of 0 means no trimming. The parameter “sample sec” controls the duration of the PCG samples accepted by the model, and it was found that setting it to 50 s resulted in the best performance. The parameter “n mels” was utilized when computing the mel spectrogram of a signal, and it determines the resolution and frequency bands of the mel spectrogram. In our study, the best performance was achieved when applying 140. When training deep learning models with limited data, the risk of overfitting is significant. To mitigate overfitting, various data augmentation techniques were employed to strengthen the model. Ultimately, the combination of mixup and cutout data augmentation techniques yielded the best results among the four data augmentation methods (mixup [13], cutout [14], FFM [36], and specaug [37]). In the context of training models with imbalanced data, the influence of majority classes often leads to a high false-negative rate. To address this issue, we applied cost-sensitive learning (CSL) during model training. Class weights were experimentally determined through a grid search ranging from 2 to 6, and the final selected value was 3. In terms of model selection, both LCNN and ResMax [38,39] architectures were considered, and while showing similar performance, the simpler structure of LCNN led us to choose it for our study. The given PCG data had three labels for the noise categories: “present”, “absent”, and “unknown”. To address the presence of the “unknown” category, which lies between “present” and “absent’,’ we developed a binary classification model. The “unknown” class was identified based on thresholds that maximize the weighted accuracy of the training data.

The sampling frequency was set to 4000 Hz, and the spectrogram was generated with a window size of 512 and a hop length of 256. The mel spectrogram was extracted using the librosa package in Python. Data augmentation techniques—namely, mixup with a coefficient of 0.7 and cutout with a coefficient of 0.8—were applied. “Cutoff” and “order” are arguments required for the butter_lowpass_filter function. We used a cutoff value of 150, and an order value of 2. The number of epochs for all models was set to 100, and the batch size was set to 64. The categorical cross-entropy was used as the loss function to minimize the loss during the training process, and the Adam optimizer was employed. The learning rate was adjusted using a sigmoid decay function with a learning rate scheduler. The initial learning rate was set to $1\times 10^{-3}$, and it gradually decreased to $1\times 10^{-5}$ according to the scheduler.

### 3.4. Model Training Details

Due to the limited amount of data provided, we addressed the data scarcity issue by using a single-instance learning structure. The dataset consisted of single or multiple signal data samples for each patient depending on the auscultation positions (MV, TV, AV, PV), and each signal data sample was labeled. During our experiments, we treated the files collected based on auscultation positions as individual samples for training our model. However, during the evaluation phase, we needed to derive a single result for each patient. Hence, it became necessary to combine individual samples based on their auscultation positions. Figure 9 visually illustrates the single-instance learning structure used in this study. During the evaluation phase, we selected the highest probability of heart murmur among the probability values calculated for each auscultation position and assigned the heart murmur label based on a certain threshold criterion.

The PhysioNet Challenge 2022 evaluated algorithmic pre-screening models for medical professionals on two fronts [9]. Firstly, weighted accuracy assigns more weight to patients with noise and abnormal results. The weighted accuracy can be calculated by referring to Table 3 and utilizing Equation (Equation 9). Secondly, a cost-based scoring metric, cost, was introduced for the clinical outcome identification task, which takes into account not only the cost of human diagnostic screening but also the costs associated with delays and missed treatments. The cost can be computed using Equation (Equation 10) with reference to Table 4. In our study, we trained the model using the training data to improve model performance and fine-tuned hyperparameters based on the important weighted accuracy metric from the validation data. As a result, other evaluation metrics’ values might be somewhat lower.

In addition to the evaluation metrics proposed in the competition, four additional metrics were computed. The area under the receiver operating characteristic curve (AUROC) represents the area under the curve of the true-positive rate with respect to the false-positive rate. The area under the precision–recall curve (AUPRC) represents the area under the curve of precision against recall. The F-measure is a metric calculated as the harmonic mean of the precision and recall. Accuracy measures how many observations, both positive and negative, were correctly classified, providing a measure of overall correctness. All the evaluation metrics, except for the cost metric, take values between 0 and 1, where values closer to 1 indicate superior model performance.
(9)$$\begin{array}{c} S_{murmur}=\frac{5m_{PP}\;+\;3m_{UU}\;+\;m_{AA}}{5\left(m_{PP}\;+\;m_{UP}\;+\;m_{AP}\right)\;+\;3\left(m_{PU}\;+\;m_{UU}\;+\;m_{AU}\right)\;+\;\left(m_{PA}\;+\;m_{UA}\;+\;m_{AA}\right)} \end{array}$$
(10)$$\begin{array}{c} S_{outcome}=\frac{5n_{TP}\;+\;n_{TN}}{5\left(n_{TP}\;+\;n_{FN}\right)\;+\;\left(n_{FP}\;+\;n_{TN}\right)} \end{array}$$

### 3.5. Experimental Results

#### 3.5.1. Use of Multi-Channel Approach and ReLCNN Model

We evaluated the performance of both the proposed multi-channel approach (depicted in Figure 5) and the proposed ReLCNN model. As a baseline, we utilized the LCNN model introduced by Lee et al. (2022) [12]. Moreover, the ReLCNN model was introduced as an architecture that incorporates the Swish activation function and adds residual blocks with the MHA mechanism to LCNN.

Table 5 presents the feature combinations applied for each experimental model ID. Table 6 and Figure 10 present the performance of the LCNN and ReLCNN models and the proposed feature combinations using six performance evaluation metrics, with Figure 10 providing a visual representation of the experimental results. PhysioNet Challenge 2022 used weighted accuracy as the evaluation metric to assess the final performance. In this study, we conducted hyperparameter tuning based on weighted accuracy to optimize the models, which might have led to relatively lower values for other evaluation metrics.

We conducted a comparison between the LCNN and ReLCNN models under the same conditions as the baseline LCNN model. The findings revealed that the LCNN model achieved a weighted accuracy of 79.6%, whereas the ReLCNN model exhibited improved performance, achieving a weighted accuracy of 80.5%. This enhancement represents an increase of approximately one percentage point. Moreover, to further improve the model’s performance, four combinations of proposed features were applied. These features included detecting S1 and S2 boundaries, detecting systolic and diastolic murmurs, using both S1 and S2 boundary detection features together with systolic and diastolic murmur features, and using the envelope as a feature without classifying S1 and S2 boundaries and murmurs. The weighted accuracies of the models with these additional features were measured as 81.1%, 82.0%, 80.8%, and 81.1%, respectively. The model incorporating S1 and S2 boundary features achieved the highest performance with a weighted accuracy of 82.0%, marking a substantial improvement from the previous model’s 79.6% weighted accuracy. These results indicate that the proposed additional features could effectively capture the characteristics of heart sound data, leading to enhanced model accuracy.

In addition, we conducted experiments by applying smoothing functions to the proposed four features. The parameter for controlling the degree of smoothing was set to 70. The models with smoothing functions exhibited weighted accuracies of 82.7%, 81.9%, 83.7%, and 83.2%, respectively. All the models with smoothing functions generally outperformed the models without smoothing. In particular, the model including systolic and diastolic murmur features achieved a weighted accuracy of 83.7%, showcasing an improvement of over four percentage points compared to the baseline model.

The application of smoothing functions in all models contributed to improving the performance of heart disease detection. Smoothing reduced noise and emphasized the dynamic characteristics of the signals, enhancing the signal-to-noise ratio in heart sound data and revealing clearer patterns related to heart diseases. Smoothing in heart sound data can play a crucial role in early detection and accurate diagnosis of heart diseases. The four proposed features demonstrated that the application of smoothing functions can effectively capture the characteristics of heart sound data and enhance model accuracy, ultimately contributing to improved detection of heart diseases.

Furthermore, to assess the impact of the PI feature, we conducted additional experiments by selecting feature combinations that exhibited the best performance in terms of weighted accuracy and cost metrics. Feature ID 11, which corresponded to a model where only the PI feature was removed from the feature ID 3 combination, achieved a weighted accuracy of 81.4% with a cost of 12,152. This indicates that, while the weighted accuracy was comparable to that of feature ID 3, the cost increased. Feature ID 12, derived from the feature ID 9 combination with the exclusion of the PI feature, demonstrated a weighted accuracy of 83.2% with a cost of 12,955. This also suggested a decrease in performance when the PI feature was not utilized.

#### 3.5.2. Optimizing Smoothing Hyperparameter (Window Size)

In Table 6, it was shown that the application of the smoothing method brings significant improvements in performance. Based on this finding, we conducted experiments to explore the optimal hyperparameters for smoothing.

We present our results in two ways. Firstly, the performance metrics for various window size (*N*) values are recorded in Table 7, and the envelope curves corresponding to different window sizes for the smoothing method are shown in Figure 11. Figure 11 displays PCG signals with the smoothing method applied, and it can be observed that, as the window size increased, the envelope became smoother. According to Table 6, features S1 and S2 (feature IDs 4 and 8) showed minimal changes after applying the smoothing method. However, for the systolic and diastolic noise features (feature IDs 5 and 9), significant differences were observed. The detection of these features was challenging due to their high variability, but the application of the smoothing method reduced the variability, contributing to increased accuracy in FHS boundary detection and, consequently, improving the performance of heart murmur detection. Furthermore, an interesting observation from Table 7 is that, for weak smoothing applications, the performance remained similar to the pre-application stage, while at window size values of around 60 to 80, both feature IDs 9 and 10 showed enhanced performance. However, excessive application of smoothing did show improvement compared to pre-application but did not lead to optimal performance.

#### 3.5.3. Impact of MHA

In this study, we conducted ablation experiments to validate the effectiveness of MHA. The ablation study involved removing or replacing MHA components based on the overall architecture described in Figure 6 while keeping the rest of the model unchanged. As evident from the results presented in Table 8, the performance significantly deteriorated when MHA components were removed. This indicates the limitations in noise detection due to various noise types. Experiments were conducted to determine the optimal number of heads, and it was found that achieving high performance required using eight heads for both feature IDs 9 and 10. This result highlights the importance of exploring noise in multiple directions. Therefore, our proposed architecture, which utilizes eight heads to preserve diverse forms of noise information, greatly enhances detection performance.

#### 3.5.4. Examining the Impact of ReLU and Swish Activation Functions

Activation function research is a crucial field that requires continuous exploration to enhance the performance of deep neural networks. Various activation functions have been proposed over time. In this study, we conducted experiments to assess the impact of the ReLU [32] and Swish [30] activation functions on the performance of deep neural networks.

The experimental results are presented in Table 9, and the activation functions were applied differently based on the best features and models from Table 6. The term “None” denotes the deep neural network model without any applied activation function, which achieved a weighted accuracy of 0.822. The model with the ReLU activation function showed a performance of 0.832, while the model with the Swish activation function achieved a weighted accuracy of 0.837.

From the experimental results, it is evident that the application of activation functions had a significant impact on performance. Although the model with the Swish activation function exhibited higher weighted accuracy, the performance difference compared to the ReLU activation function was not substantial. This suggests that the proposed ReLCNN model did not suffer from vanishing gradients due to its relatively shallow depth, leading to similar performances with both activation functions.

## 4. Discussion

### 4.1. Limitations

This study has several limitations. It is crucial to clearly understand and acknowledge these limitations for a proper interpretation of our research findings. Below, we provide a summary of the limitations of our study:Insufficient data: The study was constrained by a limited dataset comprising 942 patients, potentially affecting the model’s generalization performance. Expanding the dataset in future research could enhance the model’s robustness;Lack of widely accepted PCG databases: The absence of universally approved PCG databases remains a significant concern, hindering precise comparisons between studies. Further validation of the proposed model and features using authenticated PCG data is necessary. Collaborative efforts within the medical community to establish standardized PCG databases would greatly benefit future research;External noise: Recorded PCG signals may contain external noise, which could impact accurate diagnosis. The development of recording equipment capable of noise reduction during data acquisition is necessary;Challenges in peak detection: Due to the substantial variability in cardiac sounds, accurately detecting peak points in PCG signals can be challenging. While current algorithms rely on ECG signals for peak detection, future research should focus on developing algorithms tailored to PCG characteristics for more precise peak detection.

### 4.2. Future Work

In future research, we may consider extending the proposed MCHeart system to similar domains, like electrocardiography and echocardiography. Additionally, it would be beneficial to experiment with various state-of-the-art techniques in model architecture, such as Transformer [34] and Conformer [40].

## 5. Conclusions

In conclusion, traditional approaches have predominantly relied on spectrogram features extracted from audio data to capture time–frequency characteristics. However, our understanding of heart signals, particularly the distinct sounds of S1 and S2 during each heartbeat, offers an avenue for more nuanced analysis. Leveraging this knowledge, we proposed the multi-channel-based heart signal processing (MCHeart) system, a heart murmur detection framework designed to address irregular heart murmurs. This system consists of three key components:Enhanced cardiac activity sound features: Our approach capitalizes on additional feature information derived from cardiac activity sounds, such as S1 and S2, obtained from PCG data. By extracting cardiac activity sound features and employing mel spectrograms, we can capture richer temporal and frequency domain characteristics, augmenting the model’s capacity for detection;Smoothing function for noise minimization: We apply a smoothing function to mitigate the impact of noise or irregularities present in heart sound signals. This preprocessing step contributes to a cleaner signal representation, enhancing the system’s ability to distinguish meaningful patterns;Residual LCNN (ReLCNN) architecture: We introduce the ReLCNN architecture, which integrates residual blocks and MHA mechanisms into the LCNN model. The incorporation of MHA facilitates diverse feature extraction from multiple perspectives, enabling parallel aggregation of distinct heart murmur characteristics.

Through the implementation of the ReLCNN model, we observed a notable improvement in performance, with the weighted accuracy increasing from 79.6% to 80.5%. By incorporating the S1 and S2 components into the ReLCNN model, the weighted accuracy further rose to 82.0%. Remarkably, the application of a smoothing function yielded the most significant improvement, resulting in an accuracy of 83.7%.

Our findings underscore the potential of integrating domain knowledge, preprocessing techniques, and advanced architectures for heart murmur detection. The MCHeart system demonstrates the value of considering specific attributes of heart sounds and harnessing advanced modeling strategies to enhance accuracy and robustness in detecting irregular heart murmurs. Further research in this direction holds promise for improving the clinical applicability and accuracy of automated heart murmur detection systems.

## Figures and Tables

**Figure 1 biology-12-01291-f001:**
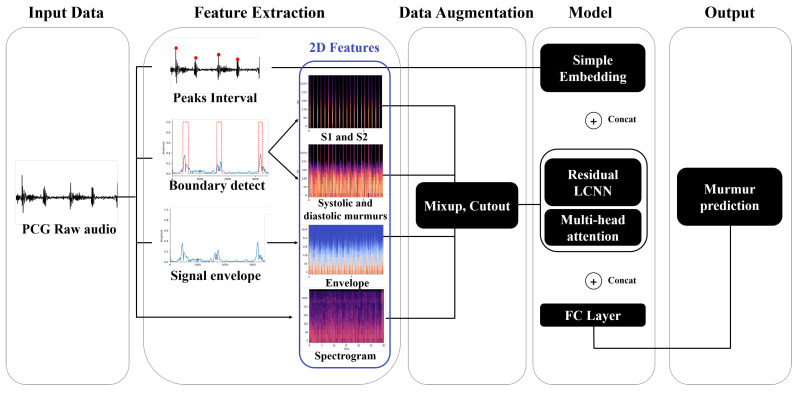
Overview of the proposed approach. We show how peak intervals, boundaries, signal envelopes, and log-mel spectrograms are extracted from phonocardiogram (PCG) raw audio data and combined with models to predict murmur (abbreviations: PCG = phonocardiogram; S1 and S2 = the two distinct sounds produced by the heart during each heartbeat; LCNN = light convolutional neural network; FC = fully connected).

**Figure 2 biology-12-01291-f002:**
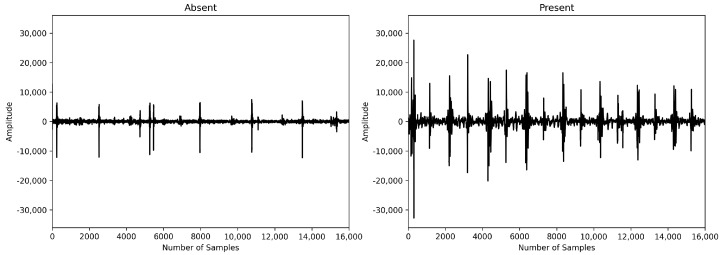
Absence vs. presence of heart murmur.

**Figure 3 biology-12-01291-f003:**
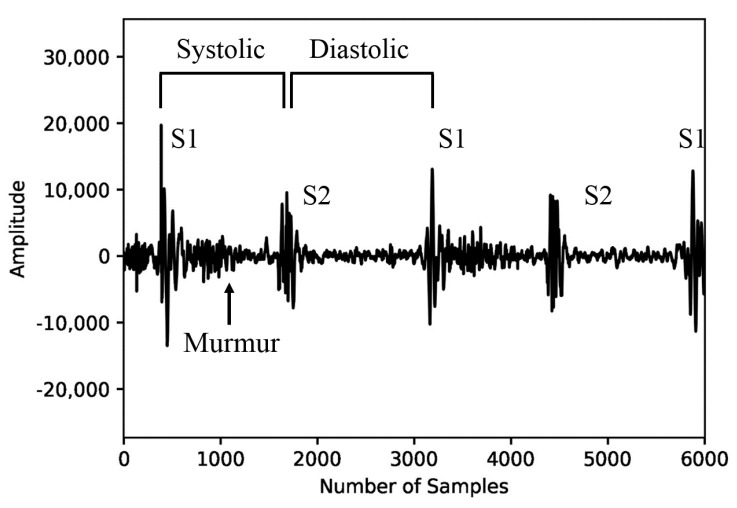
Fundamental heart sounds and heart murmurs.

**Figure 4 biology-12-01291-f004:**
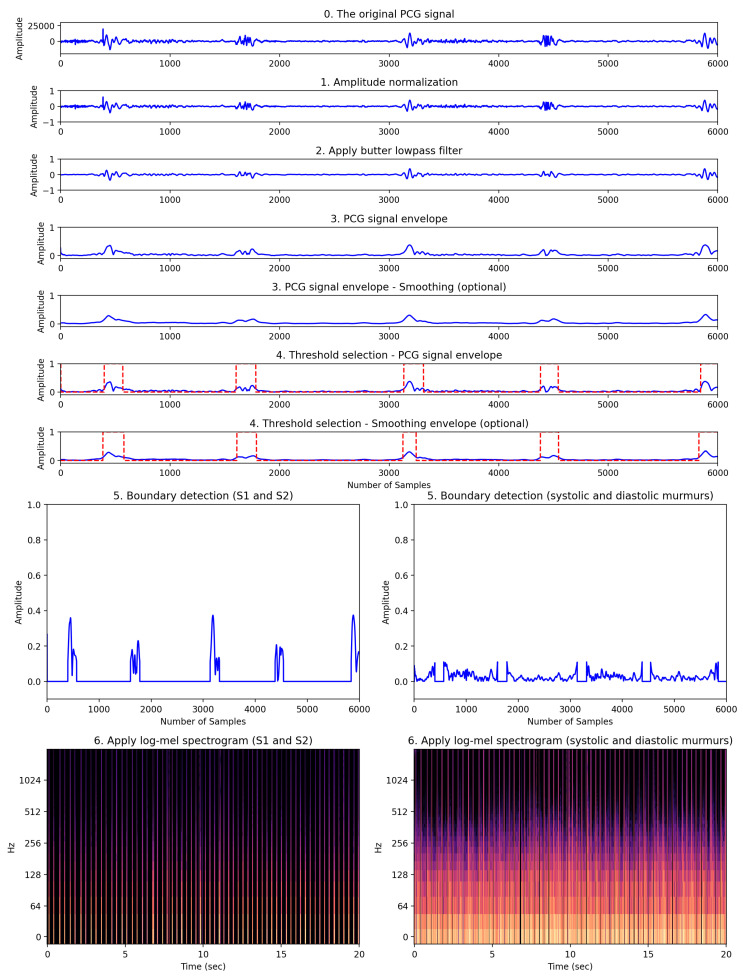
Our proposed heart feature extraction framework. This framework consists of a total of six stages visually depicting how the original PCG signal transforms through each stage.

**Figure 6 biology-12-01291-f006:**
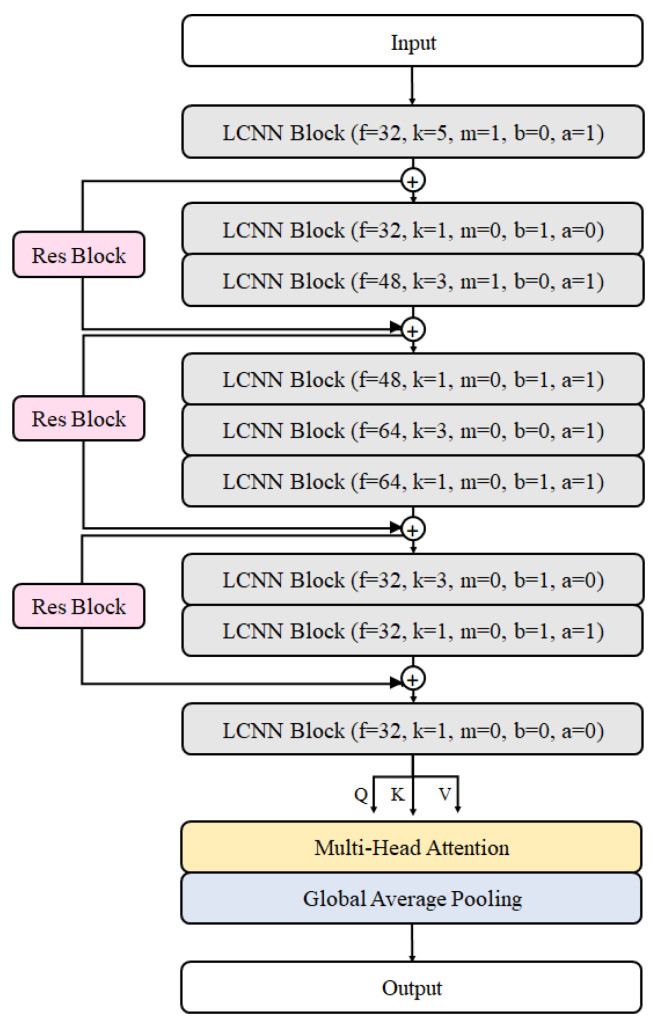
Proposed model architecture.

**Figure 7 biology-12-01291-f007:**
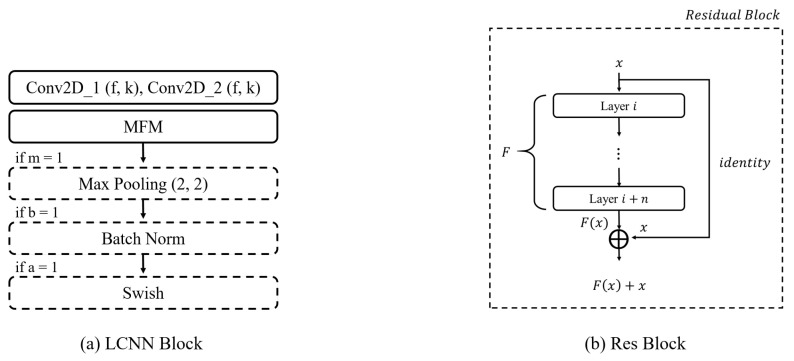
Model blocks. (**a**) LCNN block architecture. It has a convolutional layer with max-feature-map activation. Next, max pooling, batch normalization, and Swish activation are applied sequentially. (**b**) Residual block. The input feature map is added to the output feature map as a residual connection.

**Figure 8 biology-12-01291-f008:**
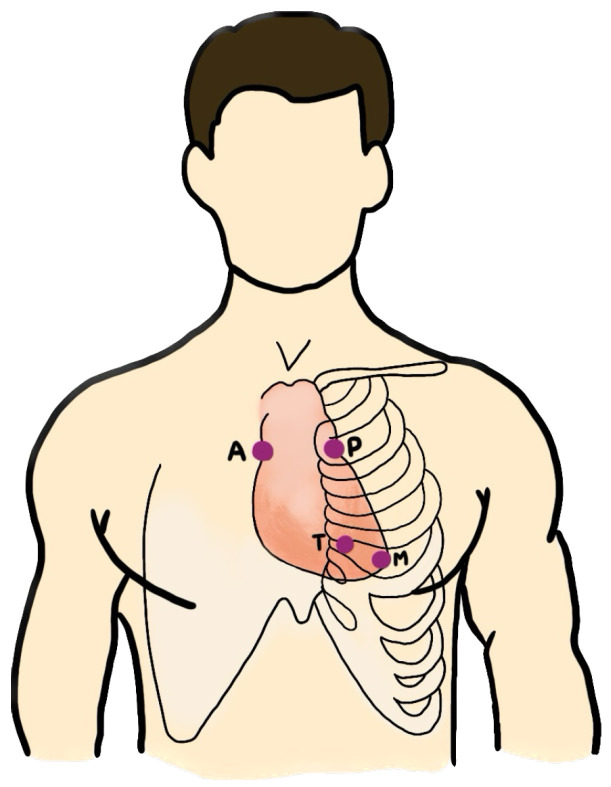
The auscultation locations: aortic valve (AV), pulmonic valve (PV), tricuspid valve (TV), and mitral valve (MV).

**Figure 9 biology-12-01291-f009:**
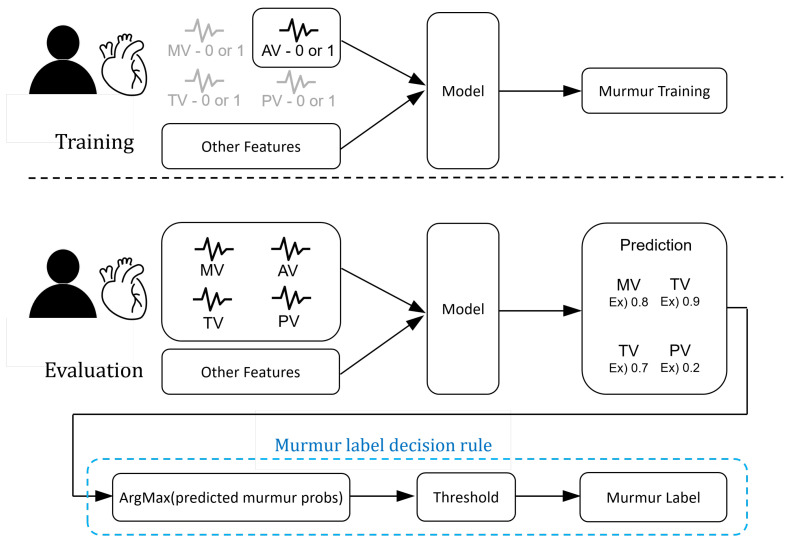
Overview of the automated murmur detection (AMD) system.

**Figure 10 biology-12-01291-f010:**
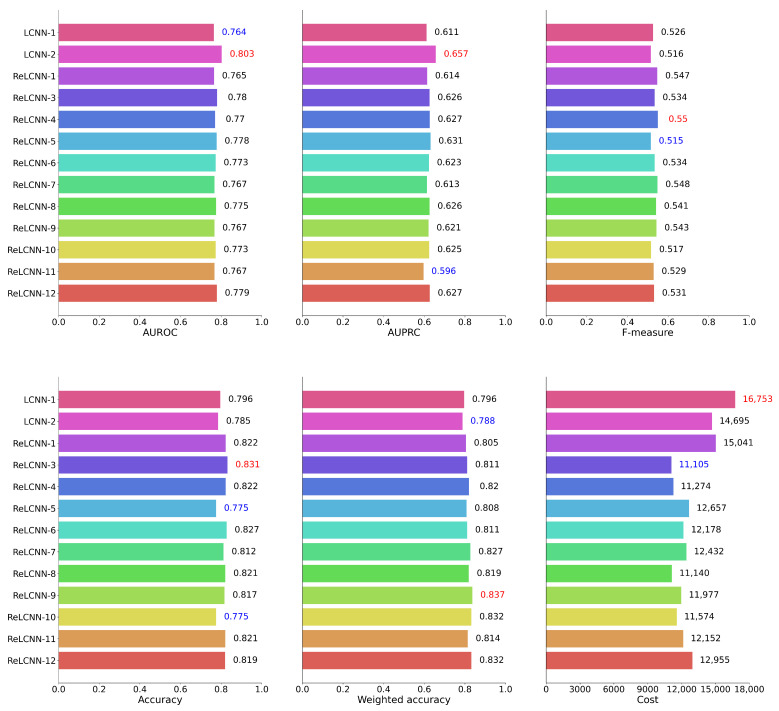
We have depicted the performance results for 12 models across six evaluation metrics in barplots. According to the weighted accuracy and cost metric used in the PhysioNet Challenge 2022, the ReLCNN-9 model and the ReLCNN-3 model exhibited the most favorable outcomes. Excluding the cost metric, the ReLCNN-9 model appears to be generally superior.

**Figure 11 biology-12-01291-f011:**
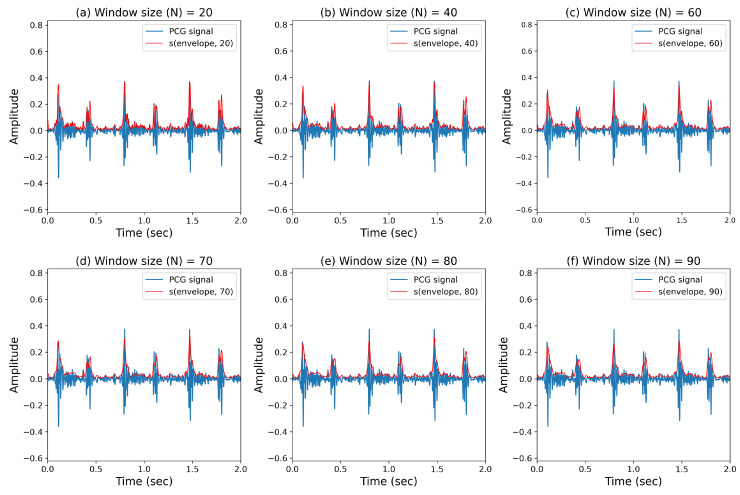
Extraction of cardiac sound envelopes based on the window size parameter from smoothing.

**Table 1 biology-12-01291-t001:** Description of training set for PhysioNet Challenge 2022.

Classification	Label	Count (%)
Murmur		
	Absent	695 (73.78%)
	Present	179 (19.00%)
	PV	62 (6.58%)
	TV	56 (5.94%)
	MV	42 (4.46%)
	AV	19 (2.02%)
	Unknown	68 (7.22%)
Total		942

**Table 2 biology-12-01291-t002:** Random search for hyperparameter optimization.

Hyperparameter	Selection	Value/Method
Spectrogram	Log-mel spectrogram	CQT, STFT, log-mel spectrogram
Trim	0	0, 2000, 4000
Sample sec	50	10, 20, 30, 40, 50
n mels	140	100, 120, 140
Data augmentation	mixup, cutout	mixup, cutout, FFM, specaug
Cost-sensitive learning (weight)	3	2, 3, 4, 5, 6
Model	LCNN	LCNN, ResMax
Inference	argmax	argmax, mean
Number of output neurons	2	2, 3

**Table 3 biology-12-01291-t003:** Murmur Metric.

		Murmur Expert		
		**Present**	**Unknown**	**Absent**
Murmur classifier	Present	$m_{PP}$	$m_{PU}$	$m_{PA}$
	Unknown	$m_{UP}$	$m_{UU}$	$m_{UA}$
	Absent	$m_{AP}$	$m_{AU}$	$m_{AA}$

**Table 4 biology-12-01291-t004:** Clinical outcome metric.

		Outcome Expert	
		**Abnormal**	**Normal**
Outcome classifier	Abnormal	$n_{TP}$	$n_{FP}$
	Normal	$n_{FN}$	$n_{TN}$

**Table 5 biology-12-01291-t005:** Feature ID allocation based on feature combinations.

Feature ID	Spec	PI	Demo	S1S2	Murmurs	Envelope	*s*(S1S2)	*s*(Murmurs)	*s*(Envelope)
1	✓	✓							
2	✓	✓	✓						
3	✓	✓		✓	✓				
4	✓	✓		✓					
5	✓	✓			✓				
6	✓	✓				✓			
7	✓	✓					✓	✓	
8	✓	✓					✓		
9	✓	✓						✓	
10	✓	✓							✓
11	✓			✓	✓				
12	✓							✓	

Abbreviation: Spec = spectrogram; PI = peak interval; Demo = demographic; S1S2 = S1 and S2; Murmurs = systolic and diastolic murmurs; *s* = smoothing function.

**Table 6 biology-12-01291-t006:** Results for performance in evaluation set.

Model	Feature ID	AUROC	AUPRC	F-Measure	Acc	Weighted Acc ↑	Cost ↓
LCNN	1	0.764	0.611	0.526	0.796	0.796	16,753
LCNN	2	0.803	0.657	0.516	0.785	0.788	14,695
ReLCNN	1	0.765	0.614	0.547	0.822	0.805	15,041
ReLCNN	3	0.780	0.626	0.534	0.831	0.811	**11,105**
ReLCNN	4	0.770	0.627	0.550	0.822	0.820	11,274
ReLCNN	5	0.778	0.631	0.515	0.775	0.808	12,657
ReLCNN	6	0.773	0.623	0.534	0.827	0.811	12,178
ReLCNN	7	0.767	0.613	0.548	0.812	0.827	12,432
ReLCNN	8	0.775	0.626	0.541	0.821	0.819	11,140
ReLCNN	9	0.767	0.621	0.543	0.817	**0.837**	11,977
ReLCNN	10	0.773	0.625	0.517	0.775	0.832	11,574
ReLCNN	11	0.767	0.596	0.529	0.821	0.814	12,152
ReLCNN	12	0.779	0.627	0.531	0.819	0.832	12,955

Abbreviation: Acc = Accuracy.

**Table 7 biology-12-01291-t007:** The weighted accuracy results when varying the window size parameter with the smoothing method.

Feature ID	Smoothing					
	20	40	60	70	80	90
9	0.804	0.811	0.825	0.837	0.836	0.819
10	0.812	0.822	0.824	0.832	0.819	0.822

**Table 8 biology-12-01291-t008:** Experimental results for MHA application and the number of heads in the ReLCNN model. The utilization of MHA yielded performance improvements in all cases based on the weighted accuracy metric, and for the cost metric, enhancements were observed in five out of six experiments. The best performance in terms of weighted accuracy was achieved when MHA was set to eight.

Feature ID	MHA	Weighted Acc ↑	Cost ↓
9	✘	0.787	11,613
	4	0.814	11,259
	8	0.837	11,977
	10	0.816	11,463
10	✘	0.795	11,872
	4	0.809	11,578
	8	0.832	11,574
	10	0.811	10,907

**Table 9 biology-12-01291-t009:** Performance comparison based on the type of activation function applied in LCNN blocks with a = 1 in the ReLCNN architecture. The Swish activation function demonstrated the most favorable results across both the weighted accuracy and cost metrics.

Feature ID	Activation	Weighted Acc ↑	Cost ↓
9	None	0.822	12,417
	ReLU	0.832	12,560
	Swish	0.837	11,977

## Data Availability

The used data are freely available at the George B. Moody PhysioNet Challenge 2022 website: https://moody-challenge.physionet.org/2022/ (accessed on 28 July 2023).

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
