# Peer review of "MCHeart: Multi-Channel-Based Heart Signal Processing Scheme for Heart Noise Detection Using Deep Learning"

_biology, 2023, doi:10.3390/biology12101291_

Round 1

Reviewer 1 Report

The work presented by the authors is very interesting and provides a very complete analysis of CNNs. However, there are some questions that we can ask, among them are the following:

The authors talk about a sampling frequency of 4000 Hz and the cutoff frequency of 150. However, they do not explain why the sampling frequency is used or if it was obtained empirically. I think it is important to mention and just as the sampling frequency is highlighted several times, to define how changing said frequency impacts the main carriers of the processed signals.

Likewise, define the robustness of the CNN, by introducing a signal with which it was not trained and a signal with some added noise or identified disturbance.

No grammatical or paragraph errors were found in the document.

Author Response

Dear reviewer,

We would like to express our sincere thanks for your time and efforts in coordinating the review process. In the revision, all the comments from you have been carefully taken into account and thoroughly addressed, with the manuscript revised accordingly.

Our response to your review is attached in the accompanying file. Thank you.

Sincerely,

Soyul Han, Woongsun Jeon, Wuming Gong, and Il-Youp Kwak

Reviewer 2 Report

The paper is interesting but could be improved. 

The abbreviations throughout the text are so many and confusing, that the  text sounds very difficult to read and understand in its present form. The Authors shoud try to avoid as much a possible any abbreviation and offer in  the text clear precise explanation of each abbreviation, when first adopted.

Line 17: .."weight"... should be ..."weighed"..?

Line 69: ..."wav"... should be .."raw".. ?

Line 73: ..."heartbeat.)." should be ..."heartbeat)." ?

Figure 1: abbreviations reported in the Figure should be explained in the legend. 

Line 89-96 appears redundant and should be deleted. 

Figure 2: try to describe differences between absent and present quadrants. 

Line 142 and following: how much high should be the amplitude of a signal to be defined as a peak ? How much the setting of the machine can influence such amplitude ? How to deal with and differentiate artifacts and noise from peaks and physiopathological murmurs? Figure 3 is a clear representation  of such a problem, since also diastole shows several small peacks: are they a noise ? how they can be differentiated from a real heart sound? Furthermore the systolic murmur of the first beat shows different components and peaks display if compared to the second beat: how to deal with these differences in practice? A precise definition of murmurs, peaks and noise should be reported. 

Figure 4. "FHS and Murmus extraction process". Please avoid FHS abbreviation. "Murmurs" insted of "Murmus" . Add some explanation to better describe what is represented in the Figure, mostly in quadrants 6. 

Lines 206-209 are not clear. Please explain.  

Lines 217-221: too many abbreviations. The text is very difficult to understand.

Figure 7: to be explained in the legend.

Line 288-291: how to deal with a selection bias with such different samples ?

Lines 292: different locations on the chest?  How these locations were precisely identified ? In fact they are very close one to the other and they can also change significantly with different chest shape and conformation, like pectus excavatum and carenatum. How to separate precisely the location as in Table 1. Please explain how to deal with reproducibiity and variability of locations and recordings of sounds and murmurs. The entire applied methodology and standardization and setting of the recordings should be better explained in detail.

Some validation of the murmurs and of the underlying diagnosis should be offered with a different technique, echocardiography as an example. 

Lines 320-325: how to deal with the risk of overfitting ?

Table 2 should be better explained in detail for a clinical cardiologist. 

Lines 416-418: to be explained in detail. 

Lines 430-433: which are the threshold amplitude and characteristics to define peaks, 1st ad 2nd heart sounds, and 3rd and 4th heart sounds, murmurs, artifacts, noise and the variablity of the same sounds during the recording ?

Lines 442-446: the performance of the model does not appear too good. Probably it could not be accepted in clinical ptactice.  A prospective validation in a new large multicenter sample population should be obtained to define accuracy and the real performance. All these aspects should be more deeply discussed in a Limitation section. The same for lines 494-496 and 525-528. 

Figure 9 is not clear and should be explained in detail in the legend.

Line 486: not clear. Please explain. 

Table 8 and 9 shoul be better explained in detail in the legend.  

A limitation sections should be introduced in the text and all limitations should be discussed in detail. 

Some editing is needed. Abbreviations should be avoided as far as possible. 

Author Response

(The authors gave the same response as above.)

Round 2

Reviewer 2 Report

The Authors made the required changes.

Minor editing required.